# Current Evidence on Immunotherapy for Gestational Trophoblastic Neoplasia (GTN)

**DOI:** 10.3390/cancers14112782

**Published:** 2022-06-03

**Authors:** Giorgia Mangili, Giulia Sabetta, Raffaella Cioffi, Emanuela Rabaiotti, Giorgio Candotti, Francesca Pella, Massimo Candiani, Alice Bergamini

**Affiliations:** 1Unit of Gynaecology and Obstetrics, IRCCS San Raffaele Scientific Institute, 20132 Milan, Italy; mangili.giorgia@hsr.it (G.M.); sabetta.giulia@hsr.it (G.S.); cioffi.raffaella@hsr.it (R.C.); rabaiotti.emanuela@hsr.it (E.R.); candotti.giorgio@hsr.it (G.C.); pella.francesca@hsr.it (F.P.); candiani.massimo@hsr.it (M.C.); 2Gynaecology and Obstetrics, Vita-Salute San Raffaele University, 20132 Milan, Italy

**Keywords:** gestational trophoblastic neoplasia, immunotherapy, PD-1/PD-L1 inhibitors, Pembrolizumab, Avelumab, Camrelizumab, Apatinib

## Abstract

**Simple Summary:**

Gestational trophoblastic neoplasia (GTN) is a rare tumor group that arises from the malignant transformation of placental tissue. Based on the evaluation of International Federation of Gynecology and Obstetrics (FIGO) anatomic staging and FIGO prognostic score, GTN is divided into low-, high-, and ultra-high-risk groups if the score obtained is less than or equal to 6, greater than 6 or greater than 12, respectively. The standard treatment is chemotherapy, using a single agent in low-risk disease and multiagent chemotherapy in high- and ultra-high-risk GTN. In chemoresistant forms of GTN, the use of immune checkpoint inhibitors, such as anti-PD-1 or anti-PD-L1/2, could represent a new therapeutic strategy. In this study, we evaluate the available evidence on immune checkpoint inhibitors for GTN treatment.

**Abstract:**

Background: Gestational trophoblastic disease includes a rare group of benign and malignant tumors derived from abnormal trophoblastic proliferation. Malignant forms are called gestational trophoblastic neoplasia (GTN) and include invasive mole, choriocarcinoma, placental site trophoblastic tumor and epithelioid trophoblastic tumor. Standard treatment of GTN is chemotherapy. The regimen of choice mainly depends on the FIGO prognostic score. Low-risk and high-risk GTN is treated with single-agent or multiagent chemotherapy, respectively. In the case of chemoresistance, immunotherapy may represent a new therapeutic strategy. Methods: Literature obtained from searches on PubMed concerning GTN and immunotherapy was reviewed. Results: Programmed cell death 1 (PD-1) and its ligands (PD-L1/2) are expressed in GTN. Published data on PD-1/PD-L1 inhibitors alone in GTN were available for 51 patients. Pembrolizumab is an anti-PD-1 inhibitor used in chemoresistant forms of GTN. In the TROPHIMMUN trial, Avelumab, a monoclonal antibody inhibiting PD-L1, showed promising results only in patients with GTN resistant to monochemotherapy. Conversely, in patients with resistance to multiagent chemotherapy, treatment with Avelumab was discontinued due to severe toxicity and disease progression. The association of Camrelizumab and Apatinib could represent a different treatment for forms of GTN refractory to polychemotherapy or for relapses. Conclusions: Anti-PD-1 or anti-PD-L1 might represent an important new treatment strategy for the management of chemoresistant/refractory GTN.

## 1. Introduction

### 1.1. Gestational Trophoblastic Disease

Gestational trophoblastic disease (GTD) is a rare pregnancy-related tumor group accounting for less than 1% of all gynecological cancers, ranging from premalignant hydatidiform mole (HM) to malignant tumors collectively appointed as gestational trophoblastic neoplasia (GTN). GTN includes several malignant forms, such as invasive mole, choriocarcinoma (CC), placental site trophoblastic tumor (PSTT) and epithelioid trophoblastic tumor (ETT) [1].

All subtypes of GTD derive from abnormal trophoblastic proliferation. HM and CC derive from villous trophoblast while PSTT and ETT arise from interstitial trophoblast [2]. The prevalence of GTD may vary according to several factors, such as geography, maternal age, previous history of molar pregnancy, reproductive history, blood groups of both partners and lifestyle (such as dietary factors and use of oral contraceptives) [2].

Hydatidiform mole represents approximately 80% of all forms of GTD. It is characterized by hydropic swelling of chorionic villi and trophoblastic proliferation. HMs are further divided into two subgroups: complete moles (CHMs) and partial moles (PHMs). The first derive from the fertilization of an empty ovum by a sperm that duplicates its DNA, producing a 46 XX androgenetic karyotype, while a partial mole is always triploid (69XXX, 69 XXY or 69XYY), and it occurs when an ovum is fertilized by two sperms. Standard treatment of hydatidiform mole requires uterine suction and curettage to be performed, ideally under ultrasound control [3,4]. Follow-up after evacuation of CHMs or PHMs requires serial serum quantitative of the β sub-unit of human chorionic gonadotropin (β-hCG) determinations every one or two weeks until three consecutive normal values are observed. Then, β-hCG levels should be dosed monthly for up to 6 months in CHMs. In PHMs, β-hCG monitoring could be discontinued after one normal value [5].

Postmolar GTN is a clinical diagnosis that occurs when hCG levels have a sustained rise or plateau. FIGO criteria for the diagnosis of postmolar GTN include: hCG plateau for four values for 3 consecutive weeks; hCG levels rise greater than 10% for three values over 2 weeks; and persistence of hCG for more than 6 months after molar evacuation [6,7,8]. The risk of postmolar GTN development after CHMs and PHMs is 15–20% and 0.5–1%, respectively [3].

GTN includes several different clinical situations, such as invasive mole, characterized by the penetration of molar villi into the myometrium or choriocarcinoma, a malignant trophoblastic tumor characterized by abnormal trophoblastic hyperplasia and anaplasia, the absence of chorionic and vascular invasion. It is often associated with metastatic lesions, most often to the lungs, brain, liver, pelvis, vagina, kidney, bowels and spleen. Usually, 50% arise from CHMs, 25% after a spontaneous miscarriage or ectopic pregnancy, and 25% of cases are associated with term delivery. The most common clinical manifestation is uterine bleeding associated with high serum hCG levels [9]. Less frequent symptoms are increased uterine volume for gestational age, hyperemesis, thyrotoxicosis and preeclampsia, related to the increased levels of β-hCG. Other symptoms are directly associated with the metastatic disease spread, such as hemoptysis, dyspnea, cough, chest pain, melena or clinical manifestations related to increased intracranial pressure from intracerebral hemorrhage, in case of brain metastases [1,3].

PSTT and ETT represent 0.2–3% of GTN cases but have the highest mortality rate [9]. PSTT is typically a tumor of the uterine corpus, and neoplastic cell infiltration is generally confined to the endometrium and myometrium, while ETT can also expand into the uterine cervix and into other localizations, such as the lung (19%). As reported by Seckl et al. in Lancet 2010, the most common symptom is represented by vaginal bleeding with a frequency varying between 31.3% and 79.4% [1]. Compared to choriocarcinoma, β-hCG levels are lower, although elevated, in 77–90% of cases [1].

PSTT and ETT can arise after full-term births or non-molar pregnancies in 95% of cases. PSTT develops after a median interval of about 3–36 months from any gestational event, while ETT develops more commonly after a mean interval of 76 months from a gestational event. The onset of these tumors after 4 years from the index pregnancy was found to be associated with a worse prognosis [10,11,12].

The pathogenesis of these tumors is poorly understood, but recent studies have revealed the epigenetic, genetic and molecular features underlining the development of GTN [13]. These new insights may help in identifying new therapeutic strategies for the treatment of chemoresistant disease.

GTD was shown to present a unique epigenomic landscape. It is important to note that the disease is more severe in CHMs, which carry the normal number of 46 human chromosomes, than in triploid PHMs. In fact, CHMs carry only paternal epigenetic signs in the methylated regions responsible for genomic imprinting, while PHMs have two paternal copies and one maternal copy. The presence of high paternal expression makes GTN highly immunogenic and vulnerable to attack in immunotherapy. This is in line with the fact that CHMs only express antigens that stimulate an alloimmune response from the maternal host [13,14]. According to the study by Szabolcsi et al., epigenetic alterations are based on DNA methylation mechanisms, which increase with the severity of GTD. Choriocarcinoma tumor cells were found to significantly overexpress DNA methyltransferase 3 beta (DNMT3B), which is the enzyme involved in de novo methylation of DNA during development. As most of the differentially expressed genes were downregulated due to the hypermethylation process, various signaling pathways, such as PI3K-Akt, ERBB2/ERBB3 and JAK-STAT, were altered, leading to, respectively, tumor cell invasion, activation of tumor cell proliferation, cell proliferation and migration mediated by cytokines and growth factors [14,15]. This potentially leads to an activation of the tumor-immune microenvironment, supporting the relevance of immunotherapy as a potential therapeutic strategy in the treatment of GTN [15,16].

### 1.2. Standard Treatment in Gestational Trophoblastic Neoplasia

Postmolar GTN requires treatment (Figure 1) if a weekly increase in β-hCG is detected at least three consecutive times over a period of at least 2 weeks, or when a plateau of β-hCG values is present for four consecutive measurements over a period of at least 3 weeks, or in the case of persistence of β-hCG 6 months after evacuation or in the case of histological diagnosis of choriocarcinoma [17]. The choice of treatment is based on the evaluation of the International Federation of Gynecology and Obstetrics (FIGO) anatomic staging of the disease (Table 1) and FIGO prognostic score (Table 2), which estimates the risk of developing chemoresistance [7]. According to these scores, tumors are classified as low-risk GTN with a FIGO prognostic score ≤ 6 and high-risk GTN if the FIGO score > 6. Ultra-high-risk disease is defined when the FIGO score is >12 [7].

Standard treatment of GTN comprises single-agent or multiagent chemotherapy for low-risk or high-risk disease, respectively [5,17].

In particular, low-risk GTN (Figure 1) is treated with methotrexate (MTX) with or without folinic acid (FA) or dactinomycin (ActD). In Europe, MTX is preferred over ActD because it causes fewer side effects (alopecia, nausea, vomiting and it is less myelosuppressive). There are several different chemotherapy regimens for MTX, but the most commonly used is the MTX eight-day regimen repeated every 14 days: a 50 mg total dose intramuscularly administered on days 1–3–5–7 with 15 mg of FA rescue given 24 h later on (days 2–4–6–8) [18,19,20]. Treatment for low-risk disease should be continued for 6 weeks after hCG normalization (three cycles of consolidation) [5]. Pharmacologically, MTX is classified as an antimetabolite due to its antagonistic effect on folic acid metabolism; in fact, it inhibits dihydrofolate reductase. Therefore, to reduce the toxic effects of MTX, it is essential to associate MTX with FA, which can enter cells through the reduced-folate transporter and be converted into tetrahydrofolate, despite the presence of MTX, reducing its cytotoxic action [21]. The most common adverse effects of MTX/FA treatment include: gastrointestinal disorders, neutropenia, anemia, elevated liver enzyme, eye disorders, such as conjunctivitis and dry eye, and fatigue. Gastrointestinal disorders account for 70% of the side effects of this chemotherapy regimen; in particular, mucositis is the most common symptom [22].

During single-agent chemotherapy, primary resistance is the increase or plateau of two consecutive β-hCG measurements, which occurs in 10–30% of patients with risk ≤ 6. Second-line monochemotherapy is generally the one that has not been used before. In fact, patients treated with MTX should have biweekly ActD if less than 1000 IU β-hCG/(actinomycin-D 10–12 µg/kg) is intravenously (IV) pushed daily for 5 days every 14 days or actinomycin-D 1.25 mg/m^2^ is IV pushed every 2 weeks) [3,23]. However, compared to MTX/FA treatment, this regimen is associated with increased toxicity, particularly hyperemesis, alopecia and risk of extravasation with local tissue damage [23,24].

If the first treatment was ActD, patients are treated with MTX with or without FA [3,23]. However, patients with risk scores equal to 5 or 6 or else with β-hCG consistently above 1000 IU/L have a 30–50% greater risk of resistant disease than those with lower prognostic scores. For these patients, switching to polychemotherapy is a reasonable option [3,21].

Patients with high-risk scores (Figure 1) are commonly treated with multiagent chemotherapy [5,17]. Multiagent chemotherapy with EMA/CO (etoposide, methothrexate, dactinomicyn, cyclophosphamide, vincristine) has a reported remission rate of 91% [25]. In cases with brain, liver or extensive lung involvement, in ultra-high-risk disease, or in all cases when high risk of bleeding is possible, induction with 1–3 cycles of low-dose EP (etoposide and cisplatin) can be considered before starting standard multiagent chemotherapy treatment. In fact, before starting EMA-CO, low-dose EP (etoposide 100 mg/m^2^ and cisplatin 20 mg/m^2^ on days 1 and 2, repeated weekly) decreases the early mortality rate from 7.2% to 0.7% [26]. In particular, in the case of brain metastases, the MTX dose in the EMA should be increased to 1 g/m^2^, alternating it weekly with CO [17]. Multiagent therapy should be continued for 6–8 weeks from normalization of β-hCG values (3–4 cycles of consolidation) [23].

In patients with risk > 6 who become refractory to the EMA/CO scheme, other regimens can be considered including EMA alternated weekly with etoposide and cisplatin (EP); paclitaxel and etoposide alternated twice weekly with paclitaxel and cisplatin (TE/TP); etoposide, ifosfamide and cisplatin (VIP) given three times a week; and bleomycin, etoposide and cisplatin (BEP) given three times weekly. Varying the limited date can be used with other regimens: ifosfamide, carboplatin and etoposide (ICE), floxuridine, actinomycin-D, etoposide and vincristine (FAEV) and gemcitabine- paclitaxel, ifosfamide and cisplatin (GEM-TIP). After treatment, serum β-hCG levels should be checked every week to determine the response until three consecutive normal values are reached; subsequent monthly monitoring for at least 6 months in low-risk GTN and 12 months for high-risk GTN patients is required. Afterwards, monitoring is performed every 6 months up to 5 years following multiagent therapy and 1 year for patients treated with single-agent chemotherapy [1,3,23,27].

Notably, these treatments are associated with short- and long-term toxicities, affecting quality of life and psychological health [28]. In addition, about 0.5–5% of patients treated with multiagent chemotherapy for high-risk GTN become chemoresistant [18]. For patients with high-risk GTN, initiation of polychemotherapy can cause rapid tumor lysis with severe bleeding, metabolic acidosis, myelosuppression and multiorgan failure, leading to death [18].

Furthermore, the main side effects of the treatment according to the EMA/CO scheme may include neutropenia, anemia, alopecia, mucositis and peripheral neuropathy. Neurotoxicity has a negative impact on the health-related quality of life of patients and may persist for months or years [25].

Moreover, etoposide is associated with an increased risk of other neoplasms, in particular leukemia, melanoma, colon cancer and breast cancer [27].

For patients with chemoresistance, high-dose chemotherapy (HDCT) with peripheral stem cell transplant can be investigated as a rescue treatment [29,30,31].

Among the main adverse events of HDCT myelosuppression, gastrointestinal toxicity, hypomagnesaemia, mucositis, peripheral neuropathy, hearing problems and infertility are the most commonly reported [32].

For chemoresistant patients, immunotherapy with immune checkpoint inhibitors might represent a new therapeutic tool [20].

PSTT and ETT, unlike other forms of GTN, have shown a poor response to chemotherapy; therefore, surgery with hysterectomy and lymph node dissection represents the treatment of choice for early-stage disease. According to recent EOTTD guidelines, adjuvant chemotherapy with platinum-based chemotherapy, high-dose chemotherapy or experimental treatment (e.g., immunotherapy) can be considered for stage I disease when antecedent pregnancy occurs more than 48 months before diagnosis, or in the case an advanced stage is reached [5,12].

## 2. Materials and Methods

In this review, manuscripts regarding GTN and immunotherapy were analyzed.

These articles were obtained from searches on Pub Med. The keywords used in the research were: GTN, choriocarcinoma, PSTT, ETT, immunotherapy, chemoresistant, Programmed cell death 1, monoclonal antibody, anti-PD-1/PD-L1 inhibitor, Pembrolizumab, TROPHIMMUN trial, Avelumab, Camrelizumab and Apatinib. The research covers the time period between January 2003 and December 2021. Published data on PD-1/PD-L1 inhibitors alone in GTN were available for 54 patients.

Of these, 12 were present in studies regarding the use of Pembrolizumab; 22 were present in the TROPHIMMUN trial (15 patients in cohort 1 and 7 in court 2); and 20 patients were in trials involving the use of Camrelizumab and Apatinib. A descriptive analysis of the single studies was carried out.

## 3. Results

### Immunotherapy and GTN

Recent evidence suggests that GTN might be an ideal candidate for immunotherapy. Indeed, several studies have demonstrated the presence of Programmed cell death 1 ligand (PD-L1) in gestational and non-gestational trophoblastic tumors, independently from FIGO score, chemoresistance or poorer clinical outcomes [33,34,35]. Programmed cell death 1 (PD-1) is a transmembrane glycoprotein that, when engaged by its ligands (PD-L1 and PD-L2), is expressed by antigen-presenting cells (APC), cancer cells and cancer cell-associated fibroblasts, inhibiting kinases that are involved in T-cell activation [33,36].

In physiological circumstances, the PD-1 and PD-L1 pathway is important for the regulation of immune responses to attenuate concomitant tissue damage from the inflammatory reaction [30]. A system by which cancer cells reduce the host immune response is up-regulation of PD-L1 and its ligation to PD-1 on antigen-specific CD8 T cells, inducing tumor progression [37,38,39]. PD-1 is expressed on the surface of activated T and B cells, regulatory T cells and natural killer (NK), cells and its primary function is performed in peripheral tissues [40].

Treatments that targets the PD-1 pathway have been investigated in various solid tumors, such as melanoma, non-small-cell lung cancer, breast cancer, colorectal cancer, renal cell carcinoma, ovarian cancer and hematologic malignancies [39,40,41,42]. The Food and Drug Administration (FDA) has approved Pembrolizumab (an anti-PD-1 inhibitor) for use in unresectable melanoma, non-small-cell lung cancer, recurrent or metastatic head and neck squamous cell cancers, classical Hodgkin lymphoma and in PD-L1-positive cervical cancer. It has a good safety profile with discontinuation rates of only 8–20% as a consequence of the toxicity [43]. Indeed, immune checkpoint inhibitors targeting PD-1 or PD-L1/2 are associated with immune-related adverse events (irAEs), which are mostly transitory and mild but can sometimes be fatal if not identified and treated. Skin manifestations, in particular rash, pruritus and mucositis, are the most common irAEs associated with these immunotherapy drugs. Other common adverse effects are: diarrhea, colitis and endocrine effects such as hypophysitis and hypothyroidism. Rare presentations involve nervous, hematopoietic, cardiovascular and urinary systems. However, if diagnosed promptly, the majority of adverse events are reversible; the use of glucocorticoids, infliximab or other agents is reasonable only in the most critical stages of diseases [44].

Recent studies have reported high expression of PD-L1 in normal placentas and in the different subtypes of gestational trophoblastic disease [33,34], comprising CHMs and CC [34], as well as in the intermediate trophoblast of PSTT and ETT. Bolze and colleagues evaluated the expression of PD-L1 in all forms of GTN, demonstrating a positivity of 80% in the choriocarcinoma specimens analyzed [33].

A new therapeutic strategy for chemoresistant forms of GTN may be drugs directed against PD-1 and its ligands (PD-L1/2). NCCN clinical practice guidelines recommend PD-1/PD-L1 inhibitors (Pembrolizumab, Nivolumab and Avelumab) as an option for the treatment of GTN resistant to chemotherapy [8].

Clinical activity in GTN cases has mainly been reported with Pembrolizumab, as summarized in Table 3. Currently, published data on PD-1/PD-L1 inhibitors alone in GTN are available for 54 patients.

Ghorani et al. have reported the outcomes of four patient treated with Pembrolizumab, all of whom had a disease resistant to multiple prior lines of therapy. This study included two patients with metastatic CC and two with metastatic PSTT and combined PSTT/ETT. Three of the four women achieved remission with anti-PD-1 after recurrence of the tumor following antecedent chemotherapy regimens. Notably, the patient who died 4 months after therapy presented mixed PSTT/ETT with strong tumor expression of PD-L1 but an absence of TILs (tumor-infiltrating lymphocytes). Moreover, tumor cells were negative for Human Leukocyte Antigen G (HLA-G). Indeed, HLA-G expression is an approved mediator of cancer immune evasion. Regarding patients with a good response to immunotherapy, the first presented with choriocarcinoma with liver and brain metastases; she was unsuccessfully treated with multiagent chemotherapy. Immunohistochemical stains showed 100% tumor expression of PD-L1 and a rich density of CD8+ cytotoxic T cells, half of which were PD-L1-positive. In addition, tumor cells were negative for HLA-A and positive for HLA-G. This patient had β-hCG normalization after four cycles of pembrolizumab. The second patient presented with a PSTT with lung, liver and brain metastases. Tumor immunohistochemistry revealed positive staining (over 90%) for PD-L1 and HLA-G and negative staining for HLA-A. This patient showed negativization of β-hCG after eight cycles of Pembrolizumab. The third patient with lung metastatic choriocarcinoma had marker normalization after only two cycles of anti-PD-L1 therapy. Immunohistochemistry revealed 100% tumor PD-L1 expression and HLA-G positivity, but tumor cells were HLA-A-negative [46].

Goldforb et al. described the case of a 50-year-old woman with 100% PD-L1-positive choriocarcinoma treated with Pembrolizumab who showed consistent disease progression following six treatment regimens: EMA/CO for 11 cycles, EMA/EP for 5 cycles, TP/TE for 7 cycles, FAEV for 4 cycles, ICE for 4 cycles and TCR105, a monoclonal antibody to endoglin, for 4 cycles). The patient underwent six cycles of Pembrolizumab, and after three cycles her β-hCG became negative. Sixteen months after the final cycle, she was disease-free [48].

Another case was reported by Clair et al. describing the clinical course in a 30-year-old woman with a metastatic gestational choriocarcinoma who, after different types of treatment, presented high PD-L1 expression and was administered Pembrolizumab, resulting in β-hCG normalization after 10 cycles. Thirty-one months after starting immunotherapy, she was still responsive [49]. In another case, Huang et al. described a chemoresistant metastatic choriocarcinoma diagnosed after a normal pregnancy. The FIGO prognostic score was 18. The patient described in the study began an induction regimen with EP followed by three cycles of EMA-CO. Before the fourth dose, β-hCG arose, and brain magnetic resonance imaging demonstrated a new cerebellar vermis lesion. Thereafter, the patient started gamma knife and EMA-EP, but after three cycles, the treatment was suspended due β-hCG increase. As a consequence of diffuse and strong membranous labeling for PD-L1 in immunohistochemistry, the patient started Pembrolizumab, obtaining a complete serologic response and a near-complete resolution of all lesions at PET/TC after two cycles [45].

Chul Choi et al. described two patients with GTN treated with a PD-1 inhibitor. A 39-year-old woman with PSTT developed metastases in the gastrointestinal tract despite multiagent chemotherapy and multiple surgeries. Immunohistochemistry of the tumor revealed 100% PD-L1, and this patient started Pembrolizumab, which resulted in normalization of β-hCG after one cycle and radiologic complete remission after four cycles. The other patient examined by Chul Choi and colleagues was a 49-year-old woman with ETT treated with numerous cytotoxic, single-agent and multiagent chemotherapies. Immunohistochemistry of the tumor showed 50% PD-L1 expression. She started treatment with Pembrolizumab, and normalization of the serological marker was observed after 11 cycles [47].

Two additional studies supported the efficacy of Pembrolizumab in the treatment of trophoblastic epithelioid cancer [3,50]. Pisani and colleagues describe the case of a previously healthy 49-year-old woman with an asymptomatic uterine mass, revealed during a routine gynecological examination. This patient underwent total abdominal hysterectomy and bilateral salpingo-oophorectomy. The histopathological and immunohistochemical analysis highlighted the presence of epithelioid trophoblastic tumor. Because genetic testing linked ETT to pregnancy in 2003, the case was considered to be high-risk, and Pembrolizumab was initiated. In this case, there was no evidence of disease recurrence for a period of 12 months [50]. The other case was described by Bell et al. and involved a 47-year-old woman diagnosed with extrauterine ETT. The patient started a chemotherapy regimen of EMA-EP for seven cycles, with partial disease response. After PD-L1 testing showed the tumor had more than 5% PD-L1 positivity, Pembrolizumab therapy was initiated. After 28 cycles, imaging demonstrated a partial response to treatment [51].

A case report by Paspalj and colleagues concerned the case of a 31-year-old woman diagnosed with choriocarcinoma following a caesarean section and with multiple lung and vaginal metastases. The patient underwent multiagent chemotherapy administration (EMA/CO and EMA/EP) without seeing any benefit. Immunohistochemical analysis showed the expression of PD-L1 in 90% of tumor cells and in 5% of immune cells. Based on these findings, she was treated with seven cycles of Pembrolizumab, achieving complete disease remission twenty-four months after the end of treatment [52].

Table 3 summarizes all cases of GTN treated with Pembrolizumab reported in the literature. Among 12 patient, 8 (66.7%) had a complete response, 3 women (25%) had a partial response, and only one patient (8.3%) had disease progression. Despite these successful cases supporting the activity of Pembrolizumab in GTN, to date no trial has been designed to test the efficacy and safety of Pembrolizumab in this disease setting. Pembrolizumab is not currently licensed for use in GTN.

Avelumab, a monoclonal antibody inhibiting PD-L1 and inducing NK-cell-mediated cytotoxicity, was studied in the setting of chemoresistant GTN in the TROPHIMMUN trial. This was an open-label multicohort, phase II trial evaluating the efficacy of this PD-L1 inhibitor in patients with chemoresistance. The trial comprised two cohorts: cohort A, dedicated to patients with GTN resistant to monochemotherapy (methotrexate or actinomycin-D) and cohort B, for patients with resistance to multiagent chemotherapy [53].

The first cohort comprised 15 patients with single-agent-resistant disease, treated with Avelumab 10 mg/kg intravenously every two weeks, until human chorionic gonadotropin (hCG) normalization. Subsequently, three further consolidation cycles were administered [53].

At enrollment, all patients were initially treated with MTX, and one patient (7%) had also received previous ActD treatment. After a median follow-up time of 25 months, 53.3% of patients (8/15) had negativization of β-hCG after a median of nine cycles, and according to the published results, one patient afterwards came pregnant. None of them had disease recurrence. A total of 46.7% of patients (seven patients) did not reach β-hCG normalization with Avelumab and were treated with subsequent chemotherapy. In detail, three women (42.3%) were treated with actinomycin-D, three (42.3%) received multiagent chemotherapy, and one (14, 3%) underwent hysterectomy. In total, 93.3% of patients had a treatment-related adverse event (TRAE), but all were grade 1 or 2. None of the patients discontinued Avelumab due to toxicity. The most common adverse events were fatigue, nausea, vomiting, infusion-related reaction and diarrhea [53]. The use of immunotherapy in patients with low-risk GTN is certainly effective, but it does not produce better results compared to traditional monochemotherapy and is much more expensive [54].

Results of the second cohort were presented at the 22nd Meeting of the European Society of Gynecological Oncology (ESGO) congress, held in Prague in October 2021. This study included seven patients in total: four with CC, one with PSTT, one with ETT and one with other forms of GTN. Only one patient had β-hCG normalization after Avelumab treatment with no subsequent relapses. Three patients experienced serious adverse events. In particular, one patient had brain hemorrhage due to arterio-veinous malformation, another patient had brain hemorrhage due to brain metastases, and one patient had a hysterectomy and salpingectomy for uterine bleeding due to disease progression. Given these results, cohort B was stopped for futility reasons.

Interestingly, a new phase I/II trial, called TROPHAMET, is currently investigating the combination of Avelumab and methotrexate in low-risk gestational trophoblastic neoplasms as a first-line treatment [55].

In the most recent version of the NCCN guidelines, Avelumab is included in the list of regimens that could potentially be effective against treatment-resistant GTN. Still, evidence is lacking regarding its efficacy in multidrug-resistant disease [8].

Another novel approach was recently investigated in this patients’ setting. A phase 2 single-arm open-label prospective study recently evaluated the activity of the combination of the PD-1 inhibitor Camrelizumab and Vascular Endothelial Growth Factor Receptor (VEGFR) inhibitor Apatinib.

Apatinib is an oral tyrosine kinase inhibitor acting by selectively binding to VEGFR2, reducing hypoxia and reprogramming the immunosuppressive tumor microenvironment. This trial included 20 patients with risk > 6, chemorefractory or relapsed disease (19 with choriocarcinoma and 1 with PSTT) who had formerly received two or more unsuccessful lines of polychemotherapy. The most common regimens prior to enrollment were EMA/CO and FAEV. Notably, 50% of enrolled patients achieved a complete response with the combination of the two drugs, and the median progression-free survival was 9.5 months. None of the patients who had a complete response to Camrelizumab and Apatinib treatment had disease recurrence after drug discontinuation. Altogether, 45% patients discontinued this treatment for disease progression and subsequently received rescue multidrug chemotherapy. A total of 77% of these patients had a complete response without recurrence. After switching to the next chemotherapy regimen, only 10% of patients died due to disease progression. Furthermore, adverse events appeared to be acceptable and manageable. Indeed, these occurred in 90% of the women in the study. The most common treatment-related grade 3 adverse events were hypertension, rash, neutropenia and leukocytopenia. Life-threatening toxicities were not observed. According to these results, this combination can be considered interesting and deserves further investigation, as it possibly represents an alternative treatment for high-risk chemorefractory or relapsed GTN [56].

## 4. Conclusions

In conclusion, immune checkpoint inhibitors targeting PD-1/PDL-1 represent a promising therapeutic strategy, potentially changing the outcomes of patients with chemorefractory GTN. These tumors are peculiar from the genomic point of view; given that its genetic inheritance is predominantly paternal, this translates into the expression of exogenous antigens of paternal origin being able to activate an immune response, making it an ideal candidate for immunotherapy [13]. The evidence derived from literature, even if scanty, supports the use of immune checkpoint inhibitors in selected settings.

Moreover, the most recent international guidelines include immunotherapy as a possible therapeutic strategy for chemotherapy-resistant GTN [5,8].

However, because of the rarity of GTN and the paucity of data in the literature, further studies are needed to provide more evidence on the efficacy of the different immune checkpoint inhibitors to identify predictive markers of response, mechanisms of resistance and to identify patients who can benefit the most from this treatment. Moreover, this will help clarifying the long-term impact of immunotherapy on fertility and the best timing for subsequent pregnancies. Considering the rarity of this disease, multicenter international collaboration on prospective trials is strongly recommended.

## Figures and Tables

**Figure 1 cancers-14-02782-f001:**
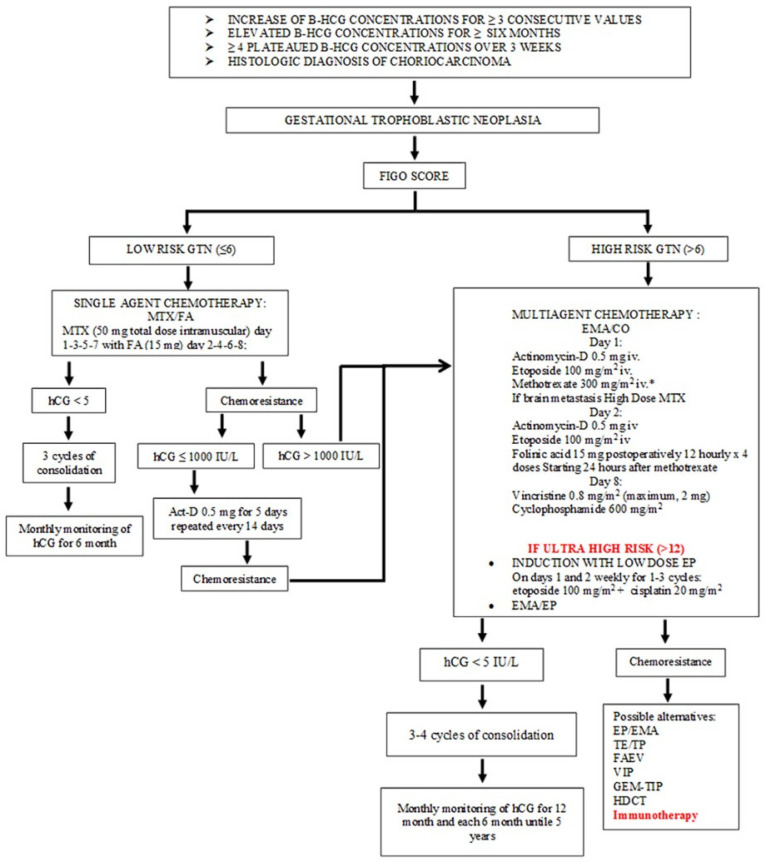
Algorithm for standard treatment of gestational trophoblastic neoplasia (adapted from Braga et al., 2019). MTX: methotrexate, ActD: actinomycin-D, EMA/CO: etoposide, methothrexate, dactinomicyn/cyclophosphamide, vincristine, EP/EMA: etoposide, cisplatin/etoposide, methothrexate, dactinomicyn, TP/TE: paclitaxel, cisplatin/paclitaxel, etoposide, VIP: etoposide, ifosfamide, cisplatin BEP: bleomycin, etoposide, cisplatin, ICE: ifosfamide, carboplatin, etoposide, FAEV: floxuridine, actinomycin-D, etoposide, vincristine, GEM-TIP: gemcitabine-paclitaxel, ifosfamide, cisplatin, HDCT: high-dose chemotherapy.

**Table 1 cancers-14-02782-t001:** Anatomic staging of GTN.

FIGO STAGING	
Stage I	GTN confined to the uterus
Stage II	GTN extends to the other genital structures
Stage III	GTN extends to the lungs, with or without genital tract involvement
Stage IV	All other distant metastases

**Table 2 cancers-14-02782-t002:** FIGO score, 2000 scoring system.

Prognostic Factors	Score 1	Score 2	Score 3	Score 4
Age	<40	> or =40		
Antecedent gestation	Mole	Abortion	Term	
Interval in months prior to end of antecedent pregnancy and start of treatment	<4	4–6	7–12	>12
Largest tumor size	<3	3–4	> or =5	
Site of metastases	Lung	Spleen, kidney	Gastrointestinal tract	Brain, liver
Number of metastases		1–4	5–8	>8
Pretreatment serum hCG (IU/L)	<10^3^	10^3^–10^4^	10^4^–10^5^	>10^5^
Previously failed chemotherapy			Single drug	Two or more drugs

**Table 3 cancers-14-02782-t003:** Review of GTN cases treated with Pembrolizumab reported in the literature (PD—progression of disease, CR—complete response, PR—partial response).

References	Tumor Type	PD-L1 Expression	Pembrolizumab Cycles to hCG Normalization	Pembrolizumab Cycles as Consolidation	**Response**
Huang et al., 2017 [45]	Choriocarcinoma	Strong	2	4	CR
Ghorani et al., 2017 [46]	Choriocarcinoma	100%	4	5	CR
	PSTT/ETT	>90%	5	0	PD
	PSTT	>90%	8	5	CR
	Choriocarcinoma	100%	2	5	CR
Chul Choi et al., 2019 [47]	PSTT	100%	1	13	CR
	ETT	50%	11	4	PR
Goldfarb et al., 2020 [48]	Choriocarcinoma	100%	3	3	CR
Clair et al., 2020 [49]	Choriocarcinoma	Strong	10	0	CR
Pisani et al., 2021 [50]	ETT	Not evaluated	Undeclared	Undeclared	CR
Bell et al., 2021 [51]	ETT	>5%	Ongoing	Ongoing	PR (Cut-off of 29 cycles)
Paspalj et al., 2021 [52]	Choriocarcinoma	>90%	4	7	CR

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
