# Peer review of "Current Evidence on Immunotherapy for Gestational Trophoblastic Neoplasia (GTN)"

_cancers, 2022, doi:10.3390/cancers14112782_

Round 1
Reviewer 1 Report
Comments and Suggestions for Authors
In this manuscript, the authors summarize GTN in a broad context from histopathological feature, diagnosis, and pharmacotherapy including immunotherapy. Although this review provides valuable information about this rare gynecological cancer for readers, the reviewer suggests some concerns that should be addressed as follows.
Major Comments
- Although the title of this manuscript is “Current evidences on immunotherapy for Gestational Trophoblastic Neoplasia (GTN)”, GTD and Standard treatment in GTN is lengthy described in the Introduction. The Reviewer understands the importance of such kinds of information, however, the authors should crisply summarize the Introduction short enough for readers to understand the background of this manuscript.
- In the Results section (3.1. Immunotherapy and GTN), the authors provide much valuable information regarding the expression profile of immune checkpoint molecules and their clinical relevance to the outcome or chemoresistance in patients with GTN, in addition to the detailed recent progress in the clinical studies for PD-1/PD-L1 inhibitors. However, given that this journal publishes high-quality articles including basic, translational, and clinical studies on all tumor types of tumors, the Reviewer suggests the authors to introduce the recent progress in the basic research for cancer immunotherapy for GTN including in vivo and also in vitro studies in the Results section. This information definitely heightens the value of this Review much further.
Minor Comments
- The resolution of Figure 1 seems to be considerably low, thus the authors should be improved the resolution quality of Figure 1.
- In the Title, “immunoterapy” seem to be typo.
- Evidence is uncountable noun. Therefore, the authors should revise the title correctly.
- In the Results Line 234, “but but” seem to be typo.
Throughout the manuscript, there are too much unnecessary spaces that should be deleted. Please check with a careful manner.
Author Response
1. Although the title of this manuscript is “Current evidences on immunotherapy for Gestational Trophoblastic Neoplasia (GTN)”, GTD and Standard treatment in GTN is lengthy described in the Introduction. The Reviewer understands the importance of such kinds of information, however, the authors should crisply summarize the Introduction short enough for readers to understand the background of this manuscript.
We thank the Reviewer for his comments. We have updated the Simple Summary (Lines 9-17, Page 1), we have also shorten the Introduction by removing some unnecessary data on epidemiology.
2. In the Results section (3.1. Immunotherapy and GTN), the authors provide much valuable information regarding the expression profile of immune checkpoint molecules and their clinical relevance to the outcome or chemoresistance in patients with GTN, in addition to the detailed recent progress in the clinical studies for PD-1/PD-L1 inhibitors. However, given that this journal publishes high-quality articles including basic, translational, and clinical studies on all tumor types of tumors, the Reviewer suggests the authors to introduce the recent progress in the basic research for cancer immunotherapy for GTN including in vivo and also in vitro studies in the Results section. This information definitely heightens the value of this Review much further.
We thank the Reviewer for this consideration. Unfortunately, to the best of our knowledge there are no available data regarding in vivo and in vitro studies on immunotherapy and GTN. We have added some recent data regarding the recent evidences on GTN genomic profiling that further support the rationale of immunotherapy in this rare disease (Lines 117-142 Page 3; Lines 463-466 Page 11; Ref. N°13-14-15-16).
3. The resolution of Figure 1 seems to be considerably low, thus the authors should be improved the resolution quality of Figure 1
We have improved the resolution of Figure 1 as suggested
4. In the Title, “immunoterapy” seem to be typo.
Typos in the title and text have been fixed
5. Evidence is uncountable noun. Therefore, the authors should revise the title correctly.
Fixed accordingly
6. In the Results Line 234, “but but” seem to be typo.
Fixed accordingly
7. Throughout the manuscript, there are too much unnecessary spaces that should be deleted. Please check with a careful manner.
Fixed accordingly

Reviewer 2 Report
The topic of this article, immune checkpoint inhibitors for the treatment of gestational trophoblastic neoplasia, has become popular and well worth discussing. However, as a review, it may lack comprehensiveness and up-to-date scientific knowledge.
- References
In the background section, the data is derailed from clinical practice. For example, regarding follow-up time for patients with hydatidiform mole and the diagnosis of post-molar, the description is inconsistent with the latest guidelines. Moreover, some references are not recent.
- Evaluation of clinical studies
Although the number of references is limited, more comprehensive data evaluation is needed. The analysis of Avelumab and the combination of Camrelizumab and Apatinib is based on only one clinical trial, respectively, and both trials are small in size, making the conclusion neither universal nor objective enough. More studies on the application of immunotherapy on GTN also need to be considered. For example, one study argues that avelumab is not recommended as a standard salvage treatment in low-risk cases (J Clin Oncol. 2020;38:4349-4350.).
In addition, one point of view is that "Avelumab showed promising results only in patients with GTN resistant to single-agent chemotherapy." This conclusion is only based on a clinical trial covering 15 patients, which is not rigorous as a conclusion. In fact, according to the latest NCCN guideline, Avelumab (Bavencio) is useful in certain circumstances in treating multiagent chemotherapy resistant GTN, and the recommended dosage is provided.
- Novelty
The content of this study is not novel, there are many similar articles. For example, one article entitled "Immune Checkpoint Inhibitors for the Treatment of Gestational Trophoblastic Neoplasia: Rationale, Effectiveness, and Future Fertility." has been published in the “Current treatment options in oncology”. There is much in common between the two articles.
- Genomic/epigenomic landscape
As gestational trophoblastic neoplasia originates mainly from gestation. The genomic/epigenomic landscape is unique and valuable, which should be associated with the sensitivity to ICI. Recently, reports of genomic analysis in trophoblastic disease is increasing. Therefore, biological characteristics should be more comprehensively discussed.
Author Response
1. References: In the background section, the data is derailed from clinical practice. For example, regarding follow-up time for patients with hydatidiform mole and the diagnosis of post-molar, the description is inconsistent with the latest guidelines. Moreover, some references are not recent.
We thank the Reviewer for this comment. As suggested, we have updated the background section with more recent references. Moreover we have modified the definition of postmolar pregnancy according to FIGO criteria (Ref. N° 7) and citing EOTTD guidelines of 2020 (Ref. N°5), and NCCN Clinical Practice Guidelines 2022 (Ref. N°8) (Lines 84-88, Page 2).
2. Evaluation of clinical studies: Although the number of references is limited, more comprehensive data evaluation is needed. The analysis of Avelumab and the combination of Camrelizumab and Apatinib is based on only one clinical trial, respectively, and both trials are small in size, making the conclusion neither universal nor objective enough. More studies on the application of immunotherapy on GTN also need to be considered. For example, one study argues that Avelumab is not recommended as a standard salvage treatment in low-risk cases (J Clin Oncol. 2020;38:4349-4350.).
The cited study was already included in our review (Ref. N° 53) and is the only study available on Avelumab in GTN. We have now included the comment on this study published on JCO, as suggested (Lines 408-410, Page 10, Ref. N° 54)
In addition, one point of view is that "Avelumab showed promising results only in patients with GTN resistant to single-agent chemotherapy." This conclusion is only based on a clinical trial covering 15 patients, which is not rigorous as a conclusion. In fact, according to the latest NCCN guideline, Avelumab (Bavencio) is useful in certain circumstances in treating multiagent chemotherapy resistant GTN, and the recommended dosage is provided.
Based on the available results we have modified the abstract, highlighting that this evidence comes from the TROPHIMMUN Trial only (Lines 29-33, Page 1). As suggested, we have cited and commented on the inclusion on Avelumab and Nivolumab for management of treatment resistant GTN according to the most recent NCCN guidelines (Lines 423-425, Page 10). Moreover, we have cited an ongoing phase I/II trial (TROPHAMET) that is currently investigating the combination of Methotrexate and Avelumab as first line treatment in low risk GTN (Lines 420-422, Page 10, Ref. 55).
3. Novelty: The content of this study is not novel, there are many similar articles. For example, one article entitled "Immune Checkpoint Inhibitors for the Treatment of Gestational Trophoblastic Neoplasia: Rationale, Effectiveness, and Future Fertility." has been published in the “Current treatment options in oncology”. There is much in common between the two articles.
The article entitled "Immune Checkpoint Inhibitors for the Treatment of Gestational Trophoblastic Neoplasm: Rationale, Effectiveness, and Future Fertility" was published online on May 5, 2022 therefore was not yet available online at the time of manuscript submission.
4. Genomic/epigenomic landscape: As gestational trophoblastic neoplasia originates mainly from gestation. The genomic/epigenomic landscape is unique and valuable, which should be associated with the sensitivity to ICI. Recently, reports of genomic analysis in trophoblastic disease is increasing. Therefore, biological characteristics should be more comprehensively discussed.
We thank the Reviewer and, as suggested, we have updated the genomic / epigenetic framework of the GTN (Lines 117-142 Page 3; Lines 463-466 Page 11; Ref. N°13-14-15-16).

Round 2
Reviewer 2 Report
The manuscript was suitably revised.